# Fire Incidents in a Mental Health Setting: Effects of Implementing Smokefree Polices and Permitting the Use of Different Types of E-Cigarettes

**DOI:** 10.3390/ijerph17238951

**Published:** 2020-12-01

**Authors:** Debbie Robson, Gilda Spaducci, Ann McNeill, Mary Yates, Melissa Wood, Sol Richardson

**Affiliations:** 1Addictions Department, Institute of Psychiatry, Psychology & Neuroscience, King’s College London, London SE5 8AF, UK; gilda.2.spaducci@kcl.ac.uk (G.S.); ann.mcneill@kcl.ac.uk (A.M.); sol.richardson@kcl.ac.uk (S.R.); 2South London & Maudsley NHS Foundation Trust, London SE5 8AZ, UK; mary.yates@slam.nhs.uk (M.Y.); melissa.wood@slam.nhs.uk (M.W.)

**Keywords:** mental health, smoking, e-cigarettes, fires

## Abstract

Comprehensive smokefree policies in health care settings can have a positive impact on patients’ smoking behaviour, but implementation is impeded by concern that surreptitious smoking may increase fire incidents. We investigated the incidence of routinely reported fire and false alarm incidents in a large mental health organisation in England over an 81-month period when different elements of a smokefree policy were implemented. We used negative binomial regression models to test associations between rates of fire and false alarm incidents and three hospital smokefree policy periods with mutual adjustment for occupied bed days: (1) an indoor policy which allowed disposable e-cigarettes; (2) a comprehensive policy which allowed disposable e-cigarettes; and (3) a comprehensive policy with all e-cigarette types allowed. We identified 90 fires and 200 false alarms. Fires decreased (incidence rate ratio (IRR): 0.35, 95% CI: 0.17–0.72, *p* = 0.004) and false alarms increased (IRR: 1.67, 95% CI: 1.02–2.76, *p* = 0.043), each by approximately two-thirds, when all e-cigarette types were allowed, after adjusting for bed occupancy and the comprehensive smokefree policy. Implementation of smokefree policies in mental health care settings that support use of all types of e-cigarettes may reduce fire risks, though measures to minimise effects of e-cigarette vapour on smoke detector systems may be needed to reduce false alarm incidents.

## 1. Introduction

Smoking prevalence among people receiving treatment for a mental health condition is two to three times greater than for those without a mental health condition [1,2]. Smoking-related illnesses are still the most common cause of early death and poor physical health for this group [3]. Comprehensive smokefree policies in health care settings, which include the prohibition of smoking on hospital premises and the provision of tobacco-dependence treatment for all patients who smoke, have been in place in parts of Europe, Australia and North America for several years [4,5,6] with varying degrees of success. In England, smoking has been prohibited in National Health Service (NHS) hospital buildings since 2007 (and mental health settings since 2008) and the implementation of comprehensive smokefree polices, including tobacco-dependence treatment pathways, have been recommended in acute, maternity and mental health settings since 2013 [4]. The National Institute of Health and Care Excellence [4] recommend hospital and community mental health services in England should provide nicotine replacement therapy (NRT), bupropion or varenicline for patients who smoke. In addition to licensed medication and behavioural support, the opportunity to switch from combustible cigarettes to e-cigarettes during a stay in a psychiatric hospital is also recommended [7]. Guidance on the use of e-cigarettes in NHS settings recommends that service provider policies and procedures should include clear directions on when and where it is appropriate to use them, how patients can access them and whether to provide them proactively to patients who smoke [8]. E-cigarettes containing nicotine are regulated by the Revised European Union Tobacco Products Directive [9] and translated into UK law through the Tobacco and Related Products Regulations 2016 [10]. The Medicines Healthcare products Regulatory Agency (MHRA) are responsible for ensuring that medicines and medical devices meet appropriate standards of safety, quality and efficacy, and are responsible for implementing the TPD. They ensure that e-cigarettes and e-liquids sold in the UK are compliant with minimum quality and safety standards (such as restricting nicotine strength to no more than 20 mg/mL) [11]. E-cigarette manufacturers can also apply to the MHRA for a medicinal licence for their products. However, there are currently no medicinally licensed e-cigarettes available to prescribe on the NHS.

Action on Smoking and Health (ASH) [12] conducted a survey of all 54 NHS mental health organisations in England in 2019. Of the 45 (83%) respondents, 37/45 (82%) reported they had a comprehensive smokefree policy in place; all reported they offered patients who smoked NRT during an inpatient stay, and 22/45 (49%) offered varenicline. The majority (41/45, 91%) allowed the use of e-cigarettes either inside hospital wards or in hospital grounds and 19 of these reported they provided free e-cigarettes to patients during an inpatient admission. Despite comprehensive smokefree policies and tobacco-dependence treatment pathways in place in the majority of NHS organisations, all survey respondents reported some level of smoking on NHS premises and patients were found smoking in their bedrooms on a weekly basis in approximately half of NHS organisations [12].

Evaluations of comprehensive smokefree policies in mental health settings demonstrate that they can have a positive impact on patients’ smoking behaviour and are associated with a reduction in physical violence towards staff and patients [13,14,15,16,17]. Although many clinicians are supportive of reducing exposure to secondhand smoke in mental health care settings, a common barrier to implementing smokefree polices is the concern that prohibiting smoking may lead to surreptitious smoking and increase the risk of fires [13,14,18]. The few studies that have evaluated the impact of the introduction of a smokefree policy in mental health hospitals on fire incidents have found no significant change in fire incidents after such a policy was implemented, relative to before. Cormac et al. [19] analysed the frequency of fire alarm incidents four months before and four months after the implementation of a comprehensive smokefree policy in a high-secure mental health hospital in England (December 2006 to July 2007) and found no fire alarm incidents attributable to surreptitious smoking. In a survey of 70 psychiatric hospitals in the USA between 2006 and 2008, the rate of fires remained unchanged in 28 hospitals where smoking was prohibited and in 42 where smoking was still allowed [20]. Voci et al. [21] evaluated the implementation of an indoor smokefree policy in a large public mental health and addiction teaching hospital in Canada between 2005 and 2008. Fifty-two fire incidents were identified 12 months prior to policy-implementation across inpatient, outpatient and emergency settings compared with 51 incidents after one year and 39 incidents after two years. These changes were not statistically significant. The same research group also evaluated changes in the rate of fire incidents following the implementation of a tobacco free policy across similar inpatient and outpatient sites between March 2014 and May 2015 [22]. Their comparison using an ANOVA test found no statistically significant change in weekly fire-related incidents at baseline compared with one-year post policy implementation.

Fire safety is a priority across the NHS in the UK. Prevention of fire incidents is important to minimise service disruption, reduce trauma, injury and expense. To date, no studies have investigated the effect of implementing smokefree polices alongside permitting the use of e-cigarettes on fire incidents in a mental health setting. The aim of this study was to investigate the association between the implementation of different elements of a comprehensive smokefree policy, which allowed for the use of e-cigarettes (initially disposable e-cigarettes for the first three years and subsequently all types of e-cigarettes), and the incidence of fires and false alarms (where the fire alarm was activated but no fire occurred) in adult wards in a large UK mental health organisation over an 81 month period spanning 1 October 2012 to 30 June 2019.

The objectives of this study were firstly to quantify the numbers of fires and false alarms attributed to patients, that occurred before and after the implementation of a comprehensive smokefree policy, and a subsequent 29 month period which allowed the use of all types of e-cigarettes, and to describe the demographic and clinical characteristics of the patients who clinical staff considered responsible for starting the fires. Secondly we sought to investigate the association between the incidence of fires and false alarms, relative to before the implementation of a comprehensive smokefree policy, and (1) the implementation of the comprehensive smokefree policy during which disposable e-cigarettes were permitted; and (2) implementation of a policy allowing all types of e-cigarettes to be used during a subsequent period.

## 2. Materials and Methods

### 2.1. Study Design, Setting and Participants

We conducted a retrospective observational study using routinely reported incident data in the South London and Maudsley (SLaM) NHS Foundation Trust, a large mental health organisation in London, UK, which provides care to a population of approximately 1.3 million people. The organisation comprises of four hospitals with approximately 50 wards and 800 beds, as well as approximately 250 community-based services including outpatients, residential and home treatment services. SLaM provides a range of clinical services including child and adolescent, adult, older adult, learning disability, forensic mental health and substance use disorder services.

An indoor smokefree policy was implemented across the organisation in 2008 and included the prohibition of smoking in inpatient wards. Smoking was allowed in ward gardens and hospital grounds with and without staff supervision [23]. A comprehensive smokefree policy was introduced on 1 October 2014. The change in policy prohibited smoking in all health care premises (hospital and community services) including both buildings and grounds. Staff-facilitated smoking such as supervised smoking breaks and escorting patients off the premises to smoke was also prohibited. The comprehensive smokefree policy was supported by a tobacco-dependence treatment care pathway, a new e-cigarette policy and a staff training programme. Treatment included the offer of NRT, varenicline or bupropion alongside behavioural support from a team of tobacco-dependence treatment advisors. Regarding the e-cigarette policy, the use of disposable e-cigarettes had been allowed in SLaM since 2012, initially only in forensic services, and from 1 October 2014 in all inpatient settings (excluding child and adolescent services). From February 2017, the e-cigarette policy was updated and the type of e-cigarettes permitted across all adult inpatient settings was extended to allow the use of reusable, rechargeable types, designed with either replaceable cartridges or pods, refillable with e-liquid by the user or modified (‘mods’) types. Heated tobacco products were not permitted in this organisation throughout the study period. 

Since 2012, e-cigarettes could be accessed by purchasing them from hospital shops or from local shops if patients had authorisation to leave the hospital site. Patients were allowed to bring in their own e-cigarettes when they were admitted to hospital provided it was within the remit of the policy. The individual use of e-cigarettes was risk assessed on a case-by-case basis and when required e-cigarettes were charged by nursing staff. Vaping was allowed in single-use bedrooms and discouraged in communal areas and during therapy sessions. Extending the choice of the type of e-cigarettes allowed was influenced by several pragmatic and clinical factors: since e-cigarettes were allowed across the whole organisation from 2014 onwards, clinicians had developed confidence and competence around their use; patients who had switched from smoking to vaping brought in their own rechargeable e-cigarettes when they were admitted to hospital, and confiscation had led to some patients relapsing back to smoking; there was a need to reduce waste from disposable e-cigarettes and promote sustainable practice. Between the implementation of the initial e-cigarette policy in 2014 and the updated version in 2017, there was emerging evidence that refillable, rechargeable e-cigarettes were the most popular type used in the wider general population [24] and that they may be more effective than disposable e-cigarettes for helping people quit smoking [25]. Further, between 2014 and 2017, there was growing evidence that e-cigarettes were substantially less harmful than smoking [26]:

The organisation’s fire alarm system involves the following: Smoke or heat detectors are fitted in all rooms of all buildings, when activated they trigger an audible alarm in the vicinity of the location where the alarm was activated, to alert building occupants. Staff in the building receive an automated message to direct them to the exact location. The fire alarm system is triggered by smoke and heat from an actual fire. Other ways the fire alarm system can be triggered when there is not a fire, includes whether aerosol spray, steam, dust and other particles come into contact with the smoke or heat detectors. Deliberate tampering with the fire alarm system can also activate the alarm. When the fire alarm is activated but there is no fire, these instances are categorised as false alarms. All staff are required to stop what they are doing and attend the vicinity of a fire alarm to establish whether there is an actual fire or false alarm. Staff then coordinate a response to ensure the safety of people in the building, this may include a disruption to the service because of evacuation or service relocation. 

For the purposes of this study, the study period was divided into three periods in which different polices were in place in SLaM mental health inpatient wards:(1)Indoor smokefree policy: the baseline period spanning October 2012 to September 2014, during which there was an indoor-only smokefree policy and disposable e-cigarettes were permitted;(2)Comprehensive smokefree policy: October 2014 to January 2017, during which the comprehensive smokefree policy was in place and disposable e-cigarettes were permitted; and(3)All e-cigarette types permitted: February 2017 to June 2019, during which the comprehensive smokefree policy continued and all e-cigarette types were permitted.

### 2.2. Data Collection and Sources

We collected information on incidents of fires and false alarms (when a fire alarm was activated but there was no fire) which occurred between 1 October 2012 and 30 June 2019. We consulted the organisation’s Fire Safety Officers and defined a fire and a false alarm according to SLaM’s Fire Policy. We categorised whether the fire incident was considered by clinical staff to be an actual fire, a false alarm, an accidental incident or a deliberate incident. Incidents were also classified by whether they occurred in an inpatient setting, community mental health centre or residential service. We extracted information from ‘Datixweb’, an online patient safety reporting system. Datixweb has previously been used in studies of patient safety incidents [16]. Staff are required to record details of incidents relating to patient safety, including fires and false alarms within 24 h. A staff member who observes the incident usually completes the online form, whilst the most senior nurse on duty is responsible for ensuring the incident is reported. The report undergoes further checks by a senior manager. Data were extracted from structured fields of Datixweb and categorised to identify date, ward and location of the incident; additional details such as the source of ignition, context and consequences of the incident were extracted from a free-text field. Data on demographic and clinical characteristics of the patients involved in fire incidents were extracted from electronic health care records. We also obtained bed occupancy data from SLaM, operationalised as monthly occupied bed days (the total of cumulative overnight patient stays) excluding patients on leave, to facilitate an investigation of potential confounding of rates of fire incidents by bed occupancy.

### 2.3. Descriptive Analysis

The three outcome variables in this study were monthly fires, false alarms, and fires and false alarms combined, attributable to patients only. We calculated the numbers of fires and false alarms recorded over the study period. The cause and ignition source of the fires; the number of injuries sustained as a result of a fire and the demographic and clinical characteristics of the patients who clinical staff believed started a fire are also reported descriptively. We graphed monthly fires, monthly false alarms and occupied bed days during the study period to describe trends in these variables. Mean monthly counts of fires, false alarms, and fires and false alarms combined were then calculated with Poisson confidence intervals for the three periods mentioned above.

### 2.4. Statistical Analysis

Correlograms and partial correlograms showed no evidence of temporal autocorrelation in the overall monthly counts of fires or false alarms; neither was there evidence of seasonal effects or a significant linear time trend. We therefore considered the assumption of serial independence to be reasonable, and initially used Poisson regression to test the association between each of the covariates mentioned above and monthly fires or false alarms. This method provides similar effect estimates to those of time series, but also a means of expressing the effects of interventions in terms of the rate ratios [27].

We hypothesised that bed days occupied per month (reflecting the total number of patients on wards) might confound any association between changes in hospital smokefree policies, change in types of e-cigarettes allowed and monthly fires attributable to patients. Four models were fitted. Models 1, 2 and 3 were unadjusted and individually tested the associations between the comprehensive smokefree policy, permitting all types of e-cigarettes, monthly bed occupancy, and each of the three outcomes. Model 4 was fully adjusted, to show the independent effect of each variable after mutual adjustment.

We used goodness-of-fit tests including the deviance statistic and Pearson’s statistic to determine whether there was overdispersion of the response variable for each outcome. The results of both tests indicated that the Poisson regression model was not appropriate due to overdispersion. We also considered fitting zero-inflated Poisson models; the likelihood ratio test indicated that use of a zero-inflated model did not improve overall goodness-of-fit. We therefore employed negative binomial regression models, which represent a generalisation of Poisson regression with an additional parameter, to loosen the restrictive assumption of Poisson regression that the conditional variance of the response is equal to the mean.

Associations between monthly incidence of fires and false alarms and implementation of the smokefree policy and change in the type of e-cigarettes permitted, in addition estimated differences in response to each additional 1000 monthly occupied bed days, were expressed in terms of incidence rate ratios (IRR). Effects for the comprehensive smokefree policy and permitting all types of e-cigarettes were shown relative to the period in which there was an indoor smokefree policy only (October 2012 to September 2014). All descriptive and statistical analyses were performed in Stata 15.

## 3. Results

### 3.1. Descriptive Analysis

Overall, between 1 October 2012 and 30 June 2019, there were 91 fires, 15 (16.5%) of which were identified as accidental, and 76 (83.5%) were considered by clinical staff to be deliberate. Ninety fires occurred on an inpatient ward, one in a community mental health centre, and none in residential settings. For consistency, we no longer refer to the one fire in the community mental health centre and focus on the inpatient settings only (for completeness of information, this fire occurred in the centre garden and was accidentally caused by a cigarette). There were 200 false alarms, of which 171/200 (85.5%) were identified as accidental and 29/200 (14.5%) were considered by staff to be deliberate. Therefore 290 inpatient fires and false alarms combined occurred in an inpatient setting during the study period.

### 3.2. Time and Location of Fires

The 90 fires in inpatient services occurred at all times of the day, though most occurred in the afternoon and evening; 16/90 (17.8%) occurred between 6AM and midday, 50/90 (55.6%) occurred between midday and 9PM and 24/90 (26.7%) occurred during the night. Most fires (79/90, 87.8%) occurred inside hospital buildings and 11/90 (12.2%) in ward gardens and hospital grounds. The most common location of a fire in hospitals buildings was bedrooms (45/79, 57%), followed by bathrooms (17/79, 21.5%). Other locations included the general areas (17/79, 25.5%) including the day room, quiet room and corridors.

### 3.3. Causes of Fires

Staff recorded the source of ignition for 54/90 (60%) fires. Smokers’ materials (e.g., tobacco cigarettes and cigarette lighters) in the context of smoking were recorded as the ignition source for 41 of 54 fires (76%) (or 45.5% of all reported fires). An e-cigarette was identified as the cause of one fire throughout the whole study period and occurred after all types of e-cigarettes were allowed. This happened whilst the e-cigarette was charging in the patient’s bedroom, which was breach of the policy. Causes of the other 12 fires included deliberately setting fire to bedsheets and/or mattresses, paper or clothing. Smokers’ materials were implicated in 7/29 (24.0%) fires during the indoor smokefree policy; 27/30 (90%) during the comprehensive smokefree policy and disposable e-cigarette period and 7/31 (23%) during the comprehensive smokefree policy and when all types of e-cigarettes were allowed period).

According to the incident reports, 76/90 (84%) of fires in inpatient settings were recorded as being started deliberately. The context and motivation were reported for 38/76 fires (50%) that were categorised as started deliberately. Almost half of these 38 fires (*n* = 18) were reportedly started because of the patient’s response to a perceived provocation, upset or threat (e.g., not being allowed to leave the ward). Seven fires were reported to have occurred as a way of enabling patients to abscond and six were reportedly started as a method of self-harm. Other reasons included wanting to harm others and boredom.

### 3.4. Causes of False Alarms

Of the 200 false alarms attributable to inpatients, the largest single cause was reported to be caused by cigarette smoke (observed or suspected by staff because of the presence or smell of tobacco smoke) (72/200, 36%). Other causes included electrical equipment (e.g., hairdryers) (15.5%), steam from showers or a deodorant aerosol (12.5%) or other non-specified reasons. Patients deliberately tampered with the fire alarm system in 14.5% of incidents. Vapour (aerosol) from e-cigarettes was observed or suspected in 30/200 (15%) incidents (29 of these occurred after all types of e-cigarettes were permitted). The Fire Service attended 61 of the false alarms; 32/71 (45%) before the comprehensive smokefree policy was implemented, 12/49 (24%) during the period when the comprehensive smokefree policy and disposable e-cigarettes were allowed organisation wide and 17/80 (21%) after all types of e-cigarettes were allowed.

### 3.5. Injuries as a Result of Fires and False Alarms

Patients and staff were injured in five fires. In one fire a patient only was injured (*n* = 1); in three fires, only staff were injured (*n* = 5) and in one fire, both patients (*n* = 1) and staff (*n* = 2) were injured. Details about the nature of the injuries, other than smoke inhalation were not recorded.

### 3.6. Demographic and Clinical Characteristics of Fires Attributed to Inpatients

Incident reports about 71 of the 90 fires (79%) included information about the patient to whom the fire had been attributed (Table 1). Seventy-one fires were attributed to 60 individual patients. Four of the 60 identified patients reportedly started two fires each and two of the 60 patients started three fires. If a patient was reported to have started more than one fire, their demographics and clinical characteristics were taken closest to the most recent time they had reportedly started a fire during the study period. Of the 60 patients with available data, the majority were men who were current smokers. There was a greater number of fires attributed to patients with a diagnosis of schizophrenia compared with other diagnostic groups. Most patients were compulsorily detained under the Mental Health Act at the time of the fire. Over half had a history of self-harm and the majority had a history of substance misuse. Almost half of all patients had a previous history of deliberate fire setting recorded in their health records.

### 3.7. Descriptive Analysis of Rate of Monthly Fires and False Alarms

The monthly mean rate of fires over the 81-month study period in an inpatient setting and attributable to patients was 1.11 (95% CI: 0.89–1.37). The monthly mean rate of false alarms was 2.47, 95% CI: 2.14–2.84), and the monthly mean rate of fires and false alarms combined was 3.58 (95% CI: 3.18–4.02). Numbers of both fires and false alarms ranged from zero to eight per month, with monthly fires peaking in April 2015 (Figure 1). Regarding the peak in April 2015, we further investigated the possible cause, including whether patient characteristics could explain this, e.g., patients with a previous history of arson may have been admitted around this time and set multiple fires. However, we only identified one patient who started two of these eight fires and they did not have a history of arson. We found no correlation between counts of monthly fires and false alarms (Pearson’s *r* = 0.030, *p* = 0.842).

Mean monthly organisation-wide occupied bed days was 21,867 (95% CI: 21,621–22,112) over the period studied. There was a gradual declining trend in monthly organisation-wide occupied bed days, from an average of 23,185 bed days per month in 2013 to 21,045 in 2018 (representing a decrease of approximately 9%). Figure 1 shows line plots of monthly fires and false alarms, and a scatter plot of occupied bed days by month across the inpatient services during the study period. The figure also illustrates the time during which each of the three policy conditions (indoor smokefree policy, comprehensive smokefree policy and organisation-wide permitted use of all e-cigarettes) were in effect.

During the period in which an indoor-only smokefree policy was in effect, there were on average 1.21 (95% CI: 0.81–1.74) fires and 2.96 (95% CI: 2.31–3.73) false alarms attributable to patients per month. After the comprehensive smokefree policy was introduced, there were 1.57 fires (95% CI: 1.14–2.11) and 1.75 (95% CI: 1.29–2.31) monthly false alarms. Following the change of the type of e-cigarettes that were permitted (from disposable only to all types), fires and false alarms occurred at a rate of 0.59 (95% CI: 0.34–0.94) and 2.75 (95% CI: 2.18–3.43) per month respectively. Figure 2 shows mean monthly fires, false alarms, and fires and false alarms combined in inpatient services, while the indoor smokefree policy, comprehensive smokefree policy and organisation-wide permitted use of e-cigarettes were in effect, with Poisson confidence intervals.

There was a decrease in the monthly rate of fires after all e-cigarette types were permitted across the organisation (February 2017 to June2019), relative to the rate of fires while the comprehensive smokefree policy and permitting only disposable e-cigarettes was in place (October 2014 to January 2017).

### 3.8. Statistical Analysis of Rate of Monthly Fires and False Alarms

The results of the negative binomial regression models (see Table 2) show that neither the comprehensive smokefree policy or bed occupancy were significantly associated with the monthly rate of fires (Models 1 and 3). We found a significant negative association between permitting all types of e-cigarettes across the whole organisation and the monthly rate of fires (Model 2 IRR: 0.42 95% CI: 0.24–0.74, *p* = 0.003), with an effect size representing a decrease of approximately half. The results of Model 4 show that the negative association between permitting all types of e-cigarettes across the whole organisation and the rate of fires remained significant after mutual adjustment for the comprehensive smokefree policy and bed occupancy (IRR: 0.35, 95% CI: 0.17–0.72, *p* = 0.004), representing an approximately two-thirds decrease in the rate of fires.

Although there was a decline in the monthly rate of false alarms following implementation of the comprehensive smokefree policy, the results of univariate Model 1 show that this was not statistically significant (IRR: 0.77, 95% CI: 0.53–1.11, *p* = 0.155). The positive univariate association between permitting all types of e-cigarettes and the monthly rate of false alarms was also non-significant (Model 2: IRR: 1.20, 95% CI: 0.83–1.71, *p* = 0.331). No significant univariate association was found between occupied bed days and the rate of false alarms. Model 4 shows, however, that after all three model variables were mutually adjusted, there was a significant negative association between implementation of the comprehensive smokefree policy and the monthly rate of false alarms (IRR: 0.62, 95% CI: 0.38–1.00, *p* = 0.049), alongside a positive association between permitting all types of e-cigarettes and the rate of false alarms, (IRR: 1.67, 95% CI: 1.02–2.76, *p* = 0.043) representing an increase in the rate of false alarms of approximately two-thirds.

No statistically significant associations were found between any of the covariates tested and monthly rates of fires and false alarms combined (Models 1–4).

## 4. Discussion

This is the first study to investigate the effect of smokefree polices and permitting the use of e-cigarettes on fire incidents in a mental health setting. Over a period of 81 months between October 2012 and June 2019, 90 fires and 200 false alarms (where a fire alarm was activated but there was no fire) were attributed to patients in the adult inpatient wards in one large mental health care organisation in the UK. The introduction of a comprehensive smokefree policy, allowing the use of disposable e-cigarettes or bed occupancy rates, did not appear to have a statistically significant effect on the monthly rate of fires. However, fires decreased by almost two-thirds when there were fewer restrictions about the type of e-cigarettes allowed, after adjusting for bed occupancy and the introduction of the comprehensive smokefree policy (Models 2 and 4). There was no statistically significant univariate association between the change in smokefree policy, change in the types of e-cigarettes allowed, or bed occupancy, on the rate of false alarms (Models 1–3). After mutual adjustment for all three variables, there was a significant reduction in the monthly rate of false alarms after the introduction of the comprehensive smokefree policy (Model 4), which only allowed disposable e-cigarettes. However, the results of the same multivariate model show that introduction of the policy permitting all types of e-cigarettes was associated with a significant increase in the monthly rate of false alarms by approximately two-thirds.

Our findings support previous studies, conducted in England and North America, which reported that introducing comprehensive smokefree policies in hospital settings does not increase the rate of fires [19,20,21,22]. Unlike previous studies that found no change in the number or rate of fire alarms [19] our study found a significant reduction in false alarms, but not fires following the implementation of a comprehensive smokefree policy. There are no other published studies in health care settings about the effect of allowing the use of e-cigarettes and the association with fire incidents to compare our findings with.

Where identified, smokers’ materials were the most common cause of fires and one fire was caused by an e-cigarette. The large difference between fires caused smoker’s materials and e-cigarettes align with those in private dwellings—between January 2015 and August 2017, the Fire Rescue Service in London, England reported 3527 fires were attributed to smokers’ materials compared with 13 attributed to e-cigarettes. Tobacco cigarette-related fires were associated with 395 injuries and 44 fatalities compared to no injuries or fatalities recorded in the same time period for e-cigarettes [28]. In the UK, the National Fire Chief’s Council UK support the notion of switching to e-cigarettes as a much safer option for one’s health and from a fire prevention perspective, compared with smoking and have published guidance on the safe use of e-cigarettes and charging within NHS settings [29]. Guidance on the safe use and charging of personal rechargeable electronic equipment including e-cigarettes on NHS premises in the UK has also been published [30]. It is notable that the one e-cigarette fire in our study period occurred when this guidance was not followed.

Among smokers in England (with and without a mental health condition), e-cigarettes are the most popular cessation aid used in a quit attempt [24]. When combined with behavioural support from trained smoking cessation practitioners, they appear to be twice as effective as NRT and behavioural support [31] and there is a small evidence base that they may be effective in reducing smoking among people with severe mental illness [32]. Data from the Smoking Toolkit Study, a repeated cross-sectional survey of approximately 1800 adults aged 16 years and over, indicate that of 2793 e-cigarette users surveyed between 2016 and 2020, who smoked or quit in the previous year, 58% of ex-smokers and 51.6% of current smokers used a refillable (tank) rechargeable e-cigarette, 16% of ex-smokers and 18% of smokers used a rechargeable prefilled cartridge e-cigarette, compared with 3% of ex-smokers and 6.5% of current smokers who used disposable e-cigarettes [24]. Refillable or prefilled rechargeable e-cigarettes have been found to be more effective at delivering nicotine and may be more effective at helping people stop smoking than disposable e-cigarettes [25,33]. In relation to our study, allowing the use of disposable, non-rechargeable e-cigarettes within the hospital setting may have reduced the motivation to surreptitiously smoke, but only to a certain extent. Once patients had a wider choice of e-cigarettes, there was a significant decrease in fires, possibly because there was a greater uptake in the use of e-cigarettes as a smoking cessation or temporary abstinence aid. Internal audits within SLaM inpatient settings suggest current e-cigarette use increased from 2%in 2013 to 14% in 2018. It is also plausible that if patients have access to more acceptable and effective e-cigarette products, that they may be more likely to use them, which may in turn reduce the desire and motivation to smoke in prohibited settings and reduce access to fire setting materials, including lighters and matches.

Tobacco smoke was the largest single cause of false alarms over the study period, although there was a statistically significant increase in false alarms triggered by e-cigarette aerosol (commonly referred to as vapour) in the study period after all types of e-cigarettes were allowed. Rechargeable e-cigarettes, with a tank that is refilled and those that can be modified, generally produce more vapour than disposable devices. Although e-cigarette vapour dissipates more quickly than tobacco smoke, if a person uses an e-cigarette continuously and generates large clouds of vapour, many standard smoke detectors such as optical devices will be sensitive to the vapour. The minority of patients who deliberately activate fire alarms, may now find e-cigarette vapour is an option to do this given that opportunities to trigger an alarm with cigarette smoke is restricted. Possible measures to mitigate this could include modifying smoke detection systems so they remain highly sensitive to smoke but are not sensitive to e-cigarette vapour [29]. The type of e-cigarette a patient brings into hospital, how they use it, the patient’s current mental state and capacity should be an integral part of routine individual risk assessments and may help identify patients at risk of deliberately activating smoke detectors. Patients should be provided with information and education about how to avoid the use of e-cigarettes near smoke detectors to reduce the likelihood of activating them. Recommending specific types of e-cigarette that are effective for smoking cessation, produce less vapour and have a lower likelihood of activating alarms should also be explored.

ASH [12] reported the majority of mental health NHS organisations in England who responded to a national survey, allowed the use of e-cigarettes in part or in all of their hospital premises. In the mental health NHS organisations that allowed vaping, all allowed disposable types, though approximately half did not allow the use of rechargeable, prefilled or refillable tank types. If the latter have potential to reduce the risk of fires, mental health NHS organisations that do not allow the use of e-cigarettes, nor allow the rechargeable, prefilled or refillable types, may wish to review their policies and practice.

There are a number of strengths and limitations of this study. This is the first study to explore the association between the impact of allowing the use of all types of e-cigarettes in a mental health setting, on fire incidents, over a long period of time, whilst controlling for other factors. However, it is limited by only including fire incidents from one NHS mental health organisation. Although this organisation is broadly representative of NHS mental health organisations across London, in terms of age, sex, ethnicity, education, and social deprivation [34], we acknowledge participants and smokefree policies might differ from the rest of the UK and other countries. While our effective model sample size of 81 monthly observations was limited, we were able to detect significant associations between changes in policies and both monthly rates of fires and false alarms. Data were unavailable on other potential confounders for these associations beyond bed occupancy rates. Further work could potentially expand on the present study by aggregating figures across multiple NHS organisations or using national-level data on fires over a longer time span and including a wider range of hospital-level control variables. However, this would require a high degree of consistency of comprehensive smokefree policy implementation, including the type of e-cigarettes allowed, as well as a high degree of consistency of reporting, and comparability of measures across different NHS organisations. We were unable to account for the vaping prevalence in the organisation, as this was not routinely recorded during the study period. A further limitation is that we were unable to retrospectively identify what type of smoke detectors were in place at each location where each fire incident occurred. Similar smoke detectors have different thresholds dependent on how they have been set, the size of the space where they are used, the amount of ventilation at the time of the incident and the amount of smoke/vapour produced. Further analysis to consider the potential impact of these issues may have been important. We also did not collect data on the financial cost to the organisation of fires and false alarms.

## 5. Conclusions

The tradition of supporting smoking and failure to provide access to tobacco-dependence treatment in mental health care settings has contributed to increased morbidity and mortality among people who smoke who use mental health services. Given the popularity of e-cigarettes, the growing evidence base for their effectiveness as cessation aids and the regulations in place [10,21,24], it is important to ensure that people who smoke and use mental health services are afforded the same opportunities as those in the wider general population to take control of their health and choose to vape instead of smoke. The results of our statistical analysis show that, after mutual adjustment for policy changes and for occupied bed days, the introduction of a comprehensive smokefree policy was associated with a reduction in the rate of fire alarms, while the policy change from a comprehensive smokefree policy which included allowing the use of disposable e-cigarettes only to a comprehensive smokefree policy which allowed the use of all types of e-cigarettes was associated with a decreased monthly rate of fires but an increased rate of false alarms. Providing care in a safe environment is an essential component of any public service. Implementation of a comprehensive smokefree policy which supports use of all types of e-cigarettes in mental health care settings has potential to reduce fire risk though measures to minimise the effects of e-cigarette vapour on smoke detector systems in these care settings may be needed to reduce false alarm incidents.

## Figures and Tables

**Figure 1 ijerph-17-08951-f001:**
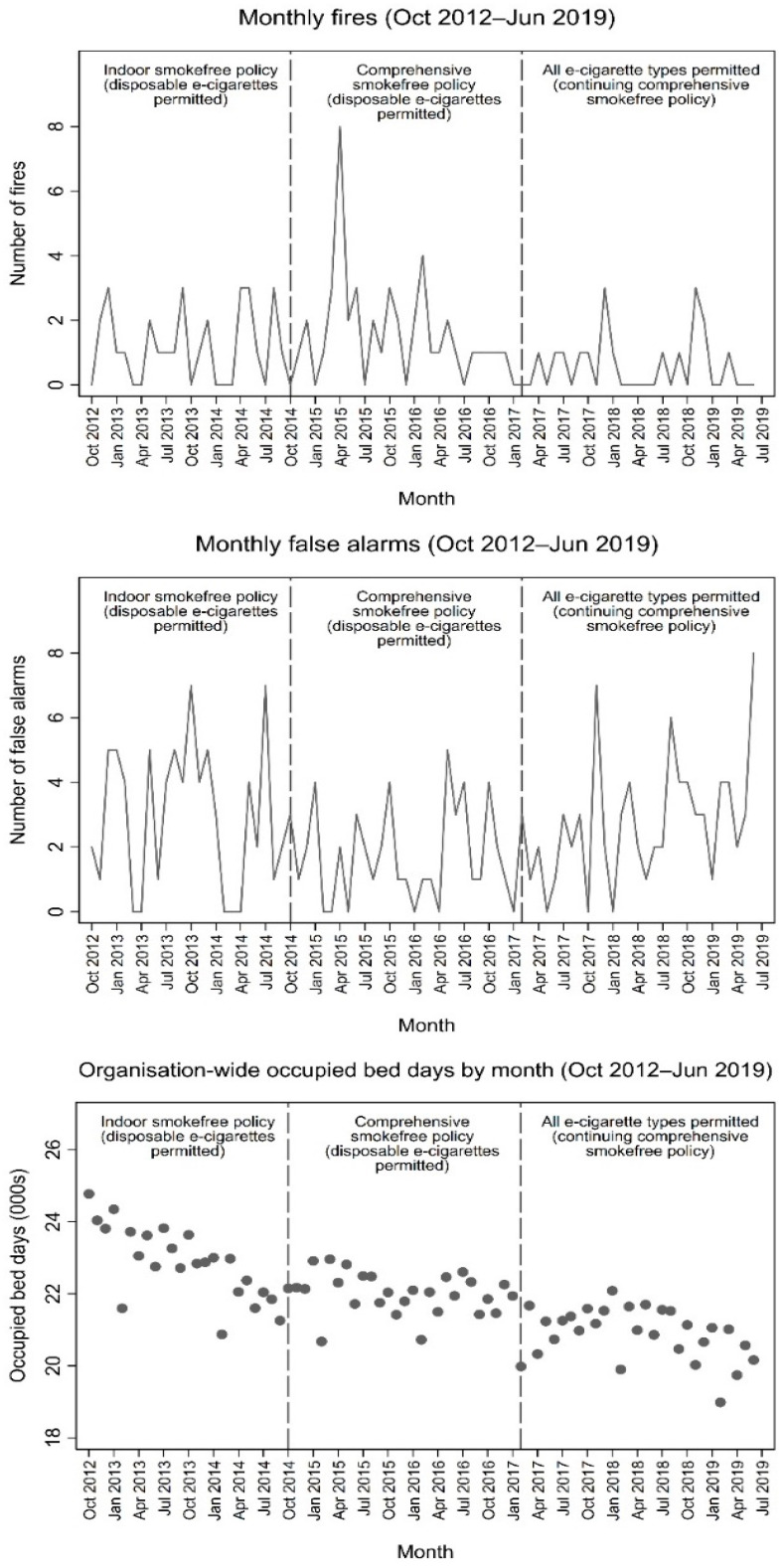
Organisation-wide monthly fires, false alarms (line graphs) and occupied bed days (scatterplots) during each policy condition (October 2012–June 2019).

**Figure 2 ijerph-17-08951-f002:**
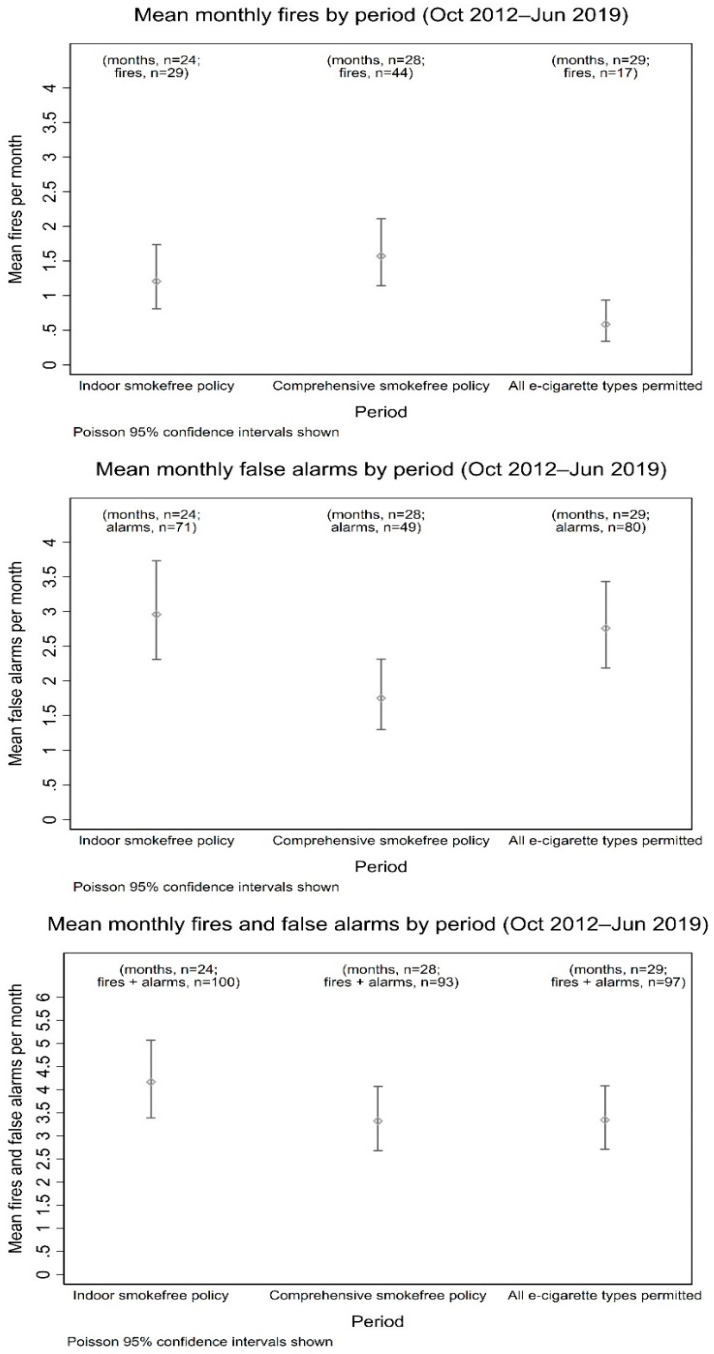
Organisation wide mean monthly fires, false alarms, combined fires and false alarms, while the indoor only smokefree policy, comprehensive smokefree policy and permitted use of all types of e-cigarettes were in effect.

**Table 1 ijerph-17-08951-t001:** Available demographic and clinical characteristics of inpatients who started fires.

Variable	Total *n* = 60 *n* (%)
Sex	
Male	42 (70.0)
Female	18 (30.0)
Mean age (sd)	33 (10.9)
Smoking status	
Smoker	50 (83.3)
Non-smoker	7 (11.7)
Missing	3 (5.0)
Diagnosis	
Schizophrenia/Schizoaffective disorder	35 (58.3)
Bipolar disorder	9 (15.0)
Personality disorder	7 (11.7)
Other	9 (15.0)
Mean number of previous admissions (sd)	5.2 (5.1)
History of previous detention under MHA	56 (93.3)
Detained under MHA at time of incident	52 (86.7)
History of self-harm	40 (66.7)
History of substance use disorder	46 (76.7)
Previous history of fire setting	26 (13.3)

**Table 2 ijerph-17-08951-t002:** Results of unadjusted and adjusted Poisson regression models for associations between smoking policies, and fires, false alarms, and fires and false alarms combined, attributable to inpatients (October 2012–June 2019).

Fires								
Variable	Model 1		Model 2		Model 3		Model 4 (Adjusted)	
	IRR (95% CI)	*p*	IRR (95% CI)	*p*	IRR (95% CI)	*p*	IRR (95% CI)	*p*
Comprehensive smokefree policy ^1^	0.89 (0.52–1.51)	0.655					1.25 (0.69–2.27)	0.462
All e-cigarettes permitted ^2^			0.42 (0.24–0.74)	0.003			0.35 (0.17–0.72)	0.004
Occupied bed days (000s)					1.18 (0.93–1.48)	0.167	0.95 (0.70–1.31)	0.771
False alarms								
Comprehensive smokefree policy ^1^	0.77 (0.53–1.11)	0.155					0.62 (0.38–1.00)	0.049
All e-cigarette types permitted ^2^			1.20 (0.83–1.71)	0.331			1.67 (1.02–2.76)	0.043
Occupied bed days (000s)					1.02 (0.88–1.19)	0.770	1.05 (0.84–1.31)	0.656
Fires and false alarms								
Comprehensive smokefree policy ^1^	0.80 (0.60–1.07)	0.134					0.81 (0.56–1.17)	0.262
All e-cigarette types permitted ^2^			0.90 (0.67–1.21)	0.486			1.02 (0.69–1.52)	0.904
Occupied bed days (000s)					1.06 (0.94–1.20)	0.323	1.01 (0.85–1.22)	0.873

^1^ Implementation of the comprehensive smokefree policy (previously only indoor areas were smokefree), binary variable (October 2012–September 2014/October 2014–June 2019). ^2^ All types of e-cigarettes permitted (previously only disposable e-cigarettes permitted), binary variable (October 2012–January 2017/February 2017–June 2019).

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
