# Peer review of "Fire Incidents in a Mental Health Setting: Effects of Implementing Smokefree Polices and Permitting the Use of Different Types of E-Cigarettes"

_ijerph, 2020, doi:10.3390/ijerph17238951_

Round 1

Reviewer 1 Report

The authors perform the effects of implementing smoke-free policies and permitting the use of different types of e-cigarettes. This study aimed to investigate the association between the implementation of different 80 elements of a comprehensive smoke-free policy. The subject seems to be in line with the International Journal of Environmental Research and Public Health. The manuscript is interesting and well organized. And, this work looks very solid and meaningful. The literature is reasonably reviewed in the manuscript. The content structure and logic of the manuscript are clear, and the main content is relatively complete. In my opinion, the presentation can be improved. The detailed comments are as follows.

1. The reviewer did not find the paper is significantly novel but only an incremental investigation which has been intensively studied before. The originality of the paper needs to be further clarified. The present form does not have sufficient results to justify the novelty. Moreover, the contribution and novelty of this paper should be clearly described in the introduction.

2. The reviewer concerns how the effects of implementing smoke-free policies and permitting the use of different types of e-cigarettes are.

3. Why did the authors concern about fire incidents in a mental health setting?

4. It is interesting in Table 2. Can you discuss more information in the text?

5. In conclusion, it may be better that the authors give your statement (standpoint) on smoke-free policies.

Author Response

Dear Reviewer, thank you for your helpful comments and your suggestions about how to improve the paper. Please find our response to your comments below.

Comment 1: The manuscript is interesting and well organized. And, this work looks very solid and meaningful. The literature is reasonably reviewed in the manuscript. The content structure and logic of the manuscript are clear, and the main content is relatively complete.

Response: Thank you.

Comment 2: The reviewer did not find the paper is significantly novel but only an incremental investigation which has been intensively studied before. The originality of the paper needs to be further clarified. The present form does not have sufficient results to justify the novelty. Moreover, the contribution and novelty of this paper should be clearly described in the introduction.

Response: Thank you. We explain in three places why this paper is novel.

At the end of paragraph 3 we state To date, no studies have investigated the effect of implementing smokefree polices alongside permitting the use of e-cigarettes on fire incidents in a mental health setting.

At the start of the discussion, we state This is the first study to investigate the effect of smokefree polices and permitting the use of e-cigarettes on fire incidents in a mental health setting.

In the final paragraph, we state, This is the first study to explore the association between the impact of allowing the use of all types of e-cigarettes in a mental health setting, on fire incidents, over a long period of time, whilst controlling for other factors.

The few studies that have investigated the effect of implementing smokefree policies on fire incidents are mentioned in paragraph 3. We are not aware of any published literature on the impact of implementing an e-cigarette policy on fire incidents in mental health settings. We would be very grateful if the reviewer could point us to papers to support the comment that this has intensively studied before.

Comment 3: The reviewer concerns how the effects of implementing smoke-free policies and permitting the use of different types of e-cigarettes are.

Response: We are not clear about the reviewer’s comment. However, we have added new text about e-cigarette regulation in the UK in the introductory section. 

E-cigarettes containing nicotine are regulated by the Revised European Union Tobacco Products Directive [9] and translated into UK law through the Tobacco and Related Products Regulations 2016 [10]. The Medicines Healthcare products Regulatory Agency (MHRA) are responsible for ensuring that medicines and medical devices meet appropriate standards of safety, quality and efficacy, and are responsible for implementing the TPD. They ensure that e-cigarettes and e-liquids sold in the UK are compliant with minimum quality and safety standards (such as restricting nicotine strength to no more than 20mg/mg/mL) [11]. E-cigarette manufacturers can also apply to the MHRA for a medicinal licence for their products, however there are currently no medicinally licensed e-cigarettes available to prescribe on the NHS’

Comment 4:  Why did the authors concern about fire incidents in a mental health setting?

Response: We explain in paragraph 3 ‘a common barrier to implementing smokefree polices is the concern that prohibiting smoking may lead to surreptitious smoking and increase the risk of fires’ supported by references.

For further clarity, we have added the following to the end of paragraph 3 in the introduction section: ‘Fire safety is a priority across the NHS in the UK. Prevention of fire incidents is important to minimise service disruption, reduce trauma, injury and expense. To date, no studies have investigated the effect of implementing smokefree polices alongside permitting the use of e-cigarettes on fire incidents in a mental health setting.’

Comment 5:  It is interesting in Table 2. Can you discuss more information in the text?

Response: An explanation of table 2 is provided in section 3.8 . We have added a few phrases to clarify this information in response your comment and other specific comments on this section from the other reviewers

Comment 6:  In conclusion, it may be better that the authors give your statement (standpoint) on smoke-free policies

Response: We have changed the two final sentences to 

‘Implementation of a comprehensive smokefree policy which supports use of all types of e-cigarettes in mental health care settings has potential to reduce risks associated with actual fires. Measures to minimise the effects of e-cigarette vapour on smoke detector systems in these care settings may reduce service disruption.

Thank you 

Reviewer 2 Report

The proposed article on “Fire incidents in a mental health setting: effects of implementing smokefree polices and permitting the use of different types of e-cigarettes” gives very comprehensive information about not well researched before topic. Overall, presented study should be evaluated as very good. I have some minor comments ad remarks, presents below.

Comments on the data collecting section:

New types of cigarettes were not taken into account – the tobacco heating systems. Those new kind of cigarettes can cause burns and can be used for setting fires. How are they treated – as normal cigarettes or the electronic ones?

In my opinion the fires caused by staff could be taken also in to account.

Comments on the discussion and comments section:

The fire risk is probably low in general. The change of smoking policy couldn’t change it much. Can the authors try to evaluate the previous and existing fire safety level in terms of risk acceptance?

Discussion lines 12 and 14 – repetition of word “however” in two consecutive sentences.

The readers could not be familiar with the fire scenario in mental health hospitals and won’t have knowledge how the fire alarm is triggered. The way that fire alarms are operating can be. Silent alarm set-up can make false alarm triggering less attractive.

Author Response

Dear Reviewer, thank you for your helpful comments and your suggestions about how to improve the paper. Please find our response to your comments below

The proposed article on “Fire incidents in a mental health setting: effects of implementing smokefree polices and permitting the use of different types of e-cigarettes” gives very comprehensive information about not well researched before topic. Overall, presented study should be evaluated as very good.

Response: Thank you

Comments on the data collecting section:

Comment 1: New types of cigarettes were not taken into account – the tobacco heating systems. Those new kind of cigarettes can cause burns and can be used for setting fires. How are they treated – as normal cigarettes or the electronic ones?

Response: As heated tobacco products are their own category, in the UK, we treat them neither as a normal cigarette or an electronic cigarette. For clarity, under section 2 materials and methods, at the end of paragraph 2 we have added ‘Heated tobacco products were not permitted in this organization throughout the study period’ 

Comment 2: In my opinion the fires caused by staff could be taken also in to account.

Response: No fires were caused by staff smoking or staff using e-cigarettes and staff are not the focus of this study.  

Comment 3: The fire risk is probably low in general. The change of smoking policy couldn’t change it much. Can the authors try to evaluate the previous and existing fire safety level in terms of risk acceptance?

Response: Evaluating risk acceptance was not the purpose of this study and we do not have data on risk acceptance. 

Comment 4: Discussion lines 12 and 14 – repetition of word “however” in two consecutive sentences.

Response: Thank you we have deleted this.

Comment 5: The readers could not be familiar with the fire scenario in mental health hospitals and won’t have knowledge how the fire alarm is triggered. The way that fire alarms are operating can be. Silent alarm set-up can make false alarm triggering less attractive.

Response: Thank you. We have added the following to section 2.1. Whilst silent alarm set-up could impact false alarm triggering, this is beyond the scope of our paper.

‘The organisation’s fire alarm system involves the following: Smoke or heat detectors are fitted in all rooms of all buildings, when activated they trigger an audible alarm in the vicinity of the location where the alarm was activated, to alert building occupants. Staff in the building receive an automated message to direct them to the exact location. The fire alarm system is triggered by smoke and heat from an actual fire. Other ways the fire alarm system can be triggered when there is not a fire, includes if aerosol spray, steam, dust and other particles come into contact with the smoke or heat detectors. Deliberate tampering with the fire alarm system can also activate the alarm. When the fire alarm is activated but there is no fire, these instances are categorised as false alarms. All staff are required to stop what they are doing and attend the vicinity of a fire alarm to establish if there is an actual fire or false alarm. Staff then coordinate a response to ensure the safety of people in the building, this may include a disruption to the service because of evacuation or service relocation’

Thank you. We hope these are our amendments following the reviewers comments address your concerns

Reviewer 3 Report

Thank you for the opportunity to review your work. Your presentation is clear and factual and provides a thorough description of your analysis methods and steps to interpret your results. I believe your manuscript provides sufficient detail such that another researcher working on an independent data set could conduct a similar analysis and assess reproducibility of your findings.

I believe the manuscript is appropriate for publication in IJERPH. I have several suggestions for the authors which, I hope, may improve the readability of the manuscript, and reflect aspects which I was confused about at first reading. As I read the paper a second time, I became less confused, but think a few of these points could be revised to enhance understanding of the manuscript.

One area that I found myself continuing to grapple with was related to the use of the terms “fires” and “false alarms.” When I first read the title and abstract, I was anticipating (clearly my inference, not necessarily intended by the authors) the article to be about fires and fire alarms related to tobacco use.

I found the introduction to be helpful in setting the tone for the article. While I work extensively with user behavior associated with tobacco use, I am less familiar with details of usage characteristics in institutional mental health settings.

As I read the manuscript the first time, I was unclear regarding incidence of fires and fire alarms. In particular, Line 83, “where the fire alarm was activated but no fire occurred” I was unclear about the nature of the fire alarms, and this question stayed in mind as I read much of the manuscript. Were these incidents of false alarms the results of a “smoke detector” automatically raising an alarm, the result of a patient initiating the alarm, or a combination of both? Perhaps a comment in the method section, clarifying that the fires and false alarms were categorized based on staff incident reports would have made reading a bit more seamless.

The statistical analysis description and justification of the negative binomial regression model was well presented.

I had to carefully read Section 3.4 to realize that fire alarms were deliberately set off 14.5% of the time, while on Line 205 it was stated right up front in Section 3.1 that 83.5% of actual fires were deliberately set.  The opening sentences on lines 204 – 208 gave me a good overall impression of the details I would be hearing about regarding fires. I feel that a similar 1 or 2 sentence summary of the false alarms, inserted between lines 209-210 would have helped set the stage for me to more readily place the subsequent results in context.

The presentation of results in Table 1, Figure 1 and Table 2 were clear and informative.

I have a clarifying question regarding Figure 2 and the statement on lines 300-302. While I do not have the underlying data, the graphical result in Figure 2 (top panel) seems to show the mean of Condition 3 outside the 95% CI of Conditions 1 and 2. Similarly, in Figure 2 (middle panel) the mean of Condition 2 appears to be outside the 95% CI of Conditions 1 and 3. Finally in Figure 2 (lower panel), it appears the mean of Condition 1 may be at the very edge of the 95% CI of Conditions 2 and 3. These observations appear to contradict the statement on Lines 300-302. Perhaps the authors might report the p values for the comparisons to make their assertions more clear.

The negative binomial regression analysis resulted in a significant association between e-cigarettes and monthly fire incidents (Lines 310-313). The analysis of false alarms (lines 324-329) asserted significance at p<0.05. It seems to me those finding are consistent with my comments regarding Figure 2 (middle panel) but are inconsistent with the earlier statement on Lines 300-302. It is unclear if I am missing some details of the two analysis approaches, or if the scale of the graphics are leading me to draw inaccurate conclusions. I think this may be a point of confusion for readers, and would encourage discussion of this point in Section 4.

The Discussion brought forward several interesting perspectives, and the Conclusions were largely supported by the data and analysis.

Author Response

Dear Reviewer, thanks for your helpful comments and your suggestions about improving the paper. Hopefully we have provided further clarity, and this now reads better.

Comment 1: Thank you for the opportunity to review your work. Your presentation is clear and factual and provides a thorough description of your analysis methods and steps to interpret your results. I believe your manuscript provides sufficient detail such that another researcher working on an independent data set could conduct a similar analysis and assess reproducibility of your findings.I believe the manuscript is appropriate for publication in IJERPH.

Response: Thank you

Comment 2: I have several suggestions for the authors which, I hope, may improve the readability of the manuscript, and reflect aspects which I was confused about at first reading. As I read the paper a second time, I became less confused, but think a few of these points could be revised to enhance understanding of the manuscript.

One area that I found myself continuing to grapple with was related to the use of the terms “fires” and “false alarms.” When I first read the title and abstract, I was anticipating (clearly my inference, not necessarily intended by the authors) the article to be about fires and fire alarms related to tobacco use.

Response: Thanks and sorry for the confusion. In relation to the title and abstract we believe it is clear that the article is about the impact of different elements of the smokefree policy on fires and false alarms, and couldn’t see how we could make this clearer. However if the reviewer has a specific recommendation on this we would happily consider this.

We have tried to make this clearer by adding the following to section 2.1

‘The organisation’s fire alarm system involves the following: Smoke or heat detectors are fitted in all rooms of all buildings, when activated they trigger an audible alarm in the vicinity of the location where the alarm was activated, to alert building occupants. Staff in the building receive an automated message to direct them to the exact location. The fire alarm system is triggered by smoke and heat from an actual fire. Other ways the fire alarm system can be triggered when there is not a fire, includes if aerosol spray, steam, dust and other particles come into contact with the smoke or heat detectors. Deliberate tampering with the fire alarm system can also activate the alarm. When the fire alarm is activated but there is no fire, these instances are categorised as false alarms. All staff are required to stop what they are doing and attend the vicinity of a fire alarm to establish if there is an actual fire or false alarm. Staff then coordinate a response to ensure the safety of people in the building, this may include a disruption to the service because of evacuation or service relocation.’

Comment 3: I found the introduction to be helpful in setting the tone for the article. While I work extensively with user behavior associated with tobacco use, I am less familiar with details of usage characteristics in institutional mental health settings.

As I read the manuscript the first time, I was unclear regarding incidence of fires and fire alarms. In particular, Line 83, “where the fire alarm was activated but no fire occurred” I was unclear about the nature of the fire alarms, and this question stayed in mind as I read much of the manuscript. Were these incidents of false alarms the results of a “smoke detector” automatically raising an alarm, the result of a patient initiating the alarm, or a combination of both? Perhaps a comment in the method section, clarifying that the fires and false alarms were categorized based on staff incident reports would have made reading a bit more seamless.

Response: Thank you, in addition to the above comment and response, we have tried to make this clearer in section 2.2

‘We collected information on incidents of fires and false alarms (when a fire alarm was activated but there was no fire) reported by staff to be attributable to patients and which occurred between 1st October 2012 and 30th June 2019. We consulted the organisation’s Fire Safety Officers and defined a fire and a false alarm according to SLaM’s Fire Policy. We categorised whether the fire incident was considered by clinical staff to be an actual fire, a false alarm, an accidental incident or a deliberate incident.

And deleted the following in the same paragraph

We also categorised whether the fire was considered by clinical staff to be accidental or deliberate.

Comment 4:  The statistical analysis description and justification of the negative binomial regression model was well presented.

Response: Thank you

Comment 5: I had to carefully read Section 3.4 to realize that fire alarms were deliberately set off 14.5% of the time,

while on Line 205 it was stated right up front in Section 3.1 that 83.5% of actual fires were deliberately set. 

The opening sentences on lines 204 – 208 gave me a good overall impression of the details I would be hearing about regarding fires. I feel that a similar 1 or 2 sentence summary of the false alarms, inserted between lines 209-210 would have helped set the stage for me to more readily place the subsequent results in context.

Response: Thank you, in addition to the above,

We have added 'There were 200 false alarms, of which 171/200 (85.5%) were identified as accidental and 29/200 (14.5%) were considered by staff to be deliberate.  Therefore 290 inpatient fires and false alarms combined occurred in an inpatient setting during the study period'.

We have taken the text ‘Patients deliberately tampered with the fire alarm system in 14.5% of incidents’ out of the sentence with the other causes of false alarms and moved it to a sentence on it’s own.

Comment 6: The presentation of results in Table 1, Figure 1 and Table 2 were clear and informative.

Response: Thank you

Comment 7: I have a clarifying question regarding Figure 2 and the statement on lines 300-302. While I do not have the underlying data, the graphical result in Figure 2 (top panel) seems to show the mean of Condition 3 outside the 95% CI of Conditions 1 and 2. Similarly, in Figure 2 (middle panel) the mean of Condition 2 appears to be outside the 95% CI of Conditions 1 and 3. Finally in Figure 2 (lower panel), it appears the mean of Condition 1 may be at the very edge of the 95% CI of Conditions 2 and 3. These observations appear to contradict the statement on Lines 300-302. Perhaps the authors might report the p values for the comparisons to make their assertions more clear.

Response: We apologise that this was not clearer and admit that the original manuscript text has the potential to cause confusion. We believe that the sentence you are referring to should be removed from the text, as it is referring to the statistical significance (or lack of) in the rate of fires between two periods when no statistical test was performed (this part of the results was just a descriptive analysis of the rates of fires and false alarms across the three periods). We have therefore deleted the sentence ‘As indicated by the overlapping confidence intervals, there was no significant change in mean monthly false alarms, and fires and false alarms combined, across the three changes in policy and e-cigarette type allowed’ to avoid possible confusion.

Comment 8: The negative binomial regression analysis resulted in a significant association between e-cigarettes and monthly fire incidents (Lines 310-313). The analysis of false alarms (lines 324-329) asserted significance at p<0.05. It seems to me those finding are consistent with my comments regarding Figure 2 (middle panel) but are inconsistent with the earlier statement on Lines 300-302. It is unclear if I am missing some details of the two analysis approaches, or if the scale of the graphics are leading me to draw inaccurate conclusions. I think this may be a point of confusion for readers, and would encourage discussion of this point in Section 4.

Response: Permitting all e-cigarettes was negatively associated with fires both before and after covariate adjustment (Models 2 and 4). Regarding false alarms, we only find significant results after covariate adjustment in Model 4 (the smokefree policy decreased the rate of false alarms but allowing all e-cigarettes increased the rate of false alarms). This is consistent with Figure 2 for the descriptive analysis (which does not show statistical tests and has overlapping Cis) but we agree that lines 300–302 are inconsistent with these results. First, as mentioned in our previous response, we deleted the sentence on lines 300–302 as it could have been misconstrued. We have also added new subsection headings and mark the division between the two parts of the analysis (emphasising that one is a descriptive analysis and the other is a statistical analysis). Finally, we checked the discussion section and find that it is consistent with the results from our statistical models—however we agree that we were somewhat unclear as to which models we were referring to and have made some minor adjustments to improve the text.

Comment 9: The Discussion brought forward several interesting perspectives, and the Conclusions were largely supported by the data and analysis.

 Response: Thank you

Thank you and we hope this addresses your helpful observations. 

Round 2

Reviewer 1 Report

After proofreading the revised manuscript, the authors have addressed my comments. The reviewer recommends the present paper for publication.